# Highly Bright Gold Nanowires Arrays for Sensitive Detection of Urea and Urease

**DOI:** 10.3390/nano12224023

**Published:** 2022-11-16

**Authors:** Yan Li, Aowei Zhao, Jieqiong Wang, Jieyu Yu, Fei Xiao, Hongcheng Sun

**Affiliations:** College of Material, Chemistry and Chemical Engineering, Key Laboratory of Organosilicon Chemistry and Material Technology, Ministry of Education, Key Laboratory of Organosilicon Material Technology of Zhejiang Province, Hangzhou Normal University, Hangzhou 311121, China

**Keywords:** gold nanowire arrays, self-assembly, enhanced fluorescence, assembly induced emission, urea and urease

## Abstract

In this work, highly fluorescent gold nanowire arrays (Au NWs) are successfully synthesized by assembling Zn^2+^ ions and non-emissive oligomeric gold-thiolate clusters using mercaptopropionic acid both as a reducing agent and a growth ligand. The synthesized Au NWs exhibited strong bluish green fluorescence with an absolute quantum yield up to 32% and possessed ultrasensitive pH stimuli-responsive performance in the range of 7.0–7.8. Based on the excellent properties of the as-prepared nanowire arrays, we developed a facile, sensitive, and selective fluorescent method for quantitative detection of urea and urease. The fabricated nanoprobe showed superior biosensing response characteristics with good linearities in the range of 0–100 μM for urea concentration and 0–12 U/L for urease activity. In addition, this fluorescent probe afforded relatively high sensitivity with the detection limit as low as 2.1 μM and 0.13 U/L for urea and urease, respectively. Urea in human urine and urease in human serum were detected with satisfied results, exhibiting a promising potential for biomedical application.

## 1. Introduction

Anomalous urea and urease levels are closely associated with the pathogenesis for many clinical symptoms in kidney disorders and hepatic disease, such as nephritic syndrome, urolithiasis, liver failure, and hepatic encephalopathy [1,2]. Urea is a nitrogen-based organic byproduct of protein metabolism that is formed in liver and excreted through the kidney in urine. Urease is a nickel-containing enzyme for urea transformation. Therefore, accurate monitoring and determination of urea and urease in human fluids are very important in clinical diagnosis of renal and liver diseases. The normal level of the urea was found to be about 2.5–7.5 mM and 155–388 mM in healthy human blood and urine, respectively [3,4]. During the recent decades, electrochemical [5,6,7,8,9], colorimetric [10,11,12], fluorescent [13,14,15], and surface-enhanced Raman scattering [16,17] approaches have been established to qualitatively and/or quantitatively detect urea and urease concentrations. Among them, fluorescence nanosensors were the most remarkable strategy due to their short response time, high sensitivity, convenience, and rapid operation [18,19]. Urea is commonly hydrolyzed in the presence of urease to produce carbon dioxide (CO_2_) and alkaline ammonia (NH_3_), and the pH of solution increased. Therefore, the concentration of urea or urease can be quantitatively monitored by the change of fluorescence of pH-sensitive biosensors. Routinely, a diverse library of fluorescent pH-dependent functional materials, such as fluorescence dyes [15,20], semiconductor quantum dots (QDs) [21,22,23,24,25], carbon-based dots (CQDs) [26,27,28], and metal nanoclusters (NCs) [29,30,31], have been employed for specially constructing urea and urease biosensors, despite their pH response behaviors are quite different.

Considering lower sensitivity resulting from the wide pH-responsive range of organic dyes, ligand stabilized colloidal QDs and CQDs, metal NCs displayed more pH-sensitive properties (ΔpH < 1) than other major nanoscale materials [30,31]. Therefore, metal NCs have constantly attracted a great deal of interest and can be used as powerful tools for pH measurement. However, the quite low fluorescence quantum yields (QY), which are generally far less than 10%, from metal NCs as pH indicators are still not sufficiently high enough to enhance their sensitivity in practical applications. For instance, *N*-acetyl-l-cysteine-capped gold nanoclusters (Au NCs), with the ultrasensitive pH stimuli-responsive performance in the range of 6.05–6.40, were used as a nanoprobe in monitoring chemical and biochemical reactions associated with pH changes [31]. The linear ranges for urea and urease were measured to be 0.055–0.55 mM and 2.2–55 U/L, respectively. Even though the Au NCs used in this work were dramatically more sensitive to pH changing, the detection linear ranges were just comparable to other reported fluorescent sensing methods because the lower QY (1.2%) of Au NCs pose a major hurdle for detecting sensitivity of biosensor.

Recently, various strategies have been developed to make weak-emitting metal NCs with highly luminescent by controlling size, tailoring structure, metal doping, and changing capped ligand [32,33,34]. Inspired by the aggregation/assembly induced emission (AIE) phenomenon, self-assembly of thiolate-capped metal NCs building blocks into desirable hierarchical architectures has proved to be a promising strategy for developing metal materials with high fluorescent emission because the formation of compact ordered aggregations could effectively restrict intramolecular vibration and rotation of the ligands to hinder the nonradiative decay [33]. For example, Xie et al. described the self-assembly of thiolate-protected Au NCs into highly ordered nanoribbons with conspicuously luminescence enhancement due to the extensive intercluster aurophilic interactions. The as-assembled nanoribbons could emit stable and red emission with a QY up to 6.2% [35]. Zhu’s group developed a novel approach based on the self-assemblies of active Cu(I)SR complexes with neutral gold atoms, leading to strong luminescence with a QY of 11.7 % [36]. Wang and co-workers reported a pH-triggered AIE methodology to fabricate D-penicillamine (DPA) coated Cu NCs by taking the advantage of noncovalent interactions of surface ligands with QY as high as 16.6% [37]. Nonetheless, the QY of these metal NC-based architectures remained relatively low compared to traditional fluorescence materials. Meanwhile, some AIE-based aggregations were carried out in organic solvent [38,39,40] or use complex and expensive capping ligands [36,41], limiting their applications in bioscience. Hence, fabrication of economic NC-based self-assembly architectures with ultrabright luminescence, superior biocompatibility, and stability will provide a versatile performance and satisfy the urgent demands in biomedical imaging, drug delivery, and biosensors in aqueous solution.

Herein, we described a metal coordination assembly strategy to develop highly fluorescent gold nanoarrays, which were further used for sensitive detection of urea and urease. Mercaptopropionic acid (MPA)-capped Au NCs could be mediated by Zn^2+^ ions through metal ion-mediated self-assembly into well defined gold nanowires (Au NWs). Fluorescence quantum yields (QY) was employed to evaluate the fluorescence efficiency of as-assembled Au NWs. Due to the metal coordination-induced fluorescence, the fluorescence of Au NWs displayed pH stimuli-responsive property, and the emission became weak, or even disappeared, when the pH increased. Because of the generation of ammonia, the pH of the urease enzymatic reaction solution could be quantitatively monitored with the change of fluorescence intensity. Such Au NWs nanomaterials can be used for the development of urea or urease biosensors. The linear relationships between fluorescence intensity versus either urea or urease concentration were built-up for quantitative detection of pending test samples. The pH-responsive range of Au NWs was within the range of physiological environment, showing potential applications in biological and medical fields.

## 2. Materials and Methods

### 2.1. Reagents and Instrumentation

All chemicals used were at least of analytical reagent grade and were used without further purification. Urease, lysozyme (Lys), bovine serum albumin (BSA), and horseradish peroxidase (HRP) were purchased from Sangon Biotech (Shanghai, China) Co., Ltd. Methionine (Met), asparagine (Asn), alanine (Ala), glutamine (Gln), glycine (Gly), proline (Pro), serine (Ser), phenylalanine (Phe), isoleucine (Ile), leucine (Leu), threonine (Thr), valine (Val), tryptophan (Trp), and lysine (Lys) were purchased from J&K Chemical LTD. HAuCl_4_·5H_2_O, NaOH, Zn(Ac)_2_, NaCl, and KCl were obtained from Aladdin reagent (Shanghai) Co., Ltd. Urea and 3-mercaptopropionic acid were purchased from Shanghai Macklin Biochemical Technology Co. Ltd. The water used in all experiments had a resistivity greater than 18 MΩ cm^−1^.

UV-vis spectra were obtained with a Shimadzu 3100 UV-VIS-NIR Recording Spectrophotometer. Dynamic Light Scattering (DLS) experiments were carried out with Malvern Instrument Zetasizer Nano ZS equipped with a He-Ne laser (633 nm, 4 mW) and an avalanche photodiode detector. Fluorescence spectra and photoluminescence lifetimes were performed on a PerkinElmer LS-55 fluorescence spectrometer and FLS 980 fluorescence spectrophotometer, respectively. Fourier transform infrared (FT-IR) spectra were conducted between 600 and 4000 cm^−1^ using a Nicolet 7000 FT-IR (Nicolet Instruments, Inc., Madison, WI, USA). The morphologies of Au NWs were measured by transmission electron microscopy (TEM, Itachi HT-7700) and scanning electron microscopy (SEM, Zeiss ultra plus and HITACHI S-4800).

### 2.2. Synthesis of Ultrabright Au NWs

The Au_4_(SC_2_H_4_COO^−^)_4_ nanoclusters (Au_4_ NCs) were synthesized based on a previously reported method [42]. Briefly, 71 μL of MPA (0.723 mmol) was mixed with 5 mL HAuCl_4_ (4 mM) under stirring for 15 min. A precipitate was formed. NaOH (1 M) was then dropped into the above mixture to bring the pH to 8–10. The precipitate was quickly dissolved to prepare Au_4_ NCs.

Besides, the amount of NaOH may be changed to further obtain water-soluble, high quantum yield Au NWs with the presence of zinc ions. After the addition of 0.75 mL zinc acetate solution (0.1 M), the mixed solution was left undisturbed in the dark for 4 h at 50 °C. The solution gradually changes to yellowish green after a few minutes. The obtained Au NWs were stored in a refrigerator for further use.

### 2.3. Fluorescence Experiments

Firstly, the pH of Au NWs was adjusted to 7.0 using 0.5 M HCl before use. For the assay of urea detection, 50 μL of urea with various concentrations were preincubated with urease at room temperature for 3 h to hold the reaction completely. The total volume of the catalytic system was 500 μL. Then, 50 μL Au NW was subsequently added to the reaction solution. After that, the solution was diluted to 2.0 mL containing the final concentration of 12 U/L urease and allowed to react for 15 min. Fluorescence measurement of urease activity was performed by incubating 50 μL of urease with various concentrations and urea for 3 h, followed by the addition of 50 μL Au NW and 1450 μL deionized water before measurement. The detection of urea and urease was conducted in water. The fluorescence emission spectra were recorded under 360 nm excitation wavelength.

### 2.4. Detection of Urea in Human Urine Sample and Urease in Serum

Human urine and serum samples were obtained from healthy volunteers with informed consent. The human urines from the same adult male were respectively collected from morning urine and after drinking about 2 L water. The biological samples were pre-treated by ultrafiltration at 8000 rpm for 10 min to precipitate the proteins, and the samples were diluted to 100 times before the urea and urease bioassay using the standard addition experiments.

## 3. Results

### 3.1. Optimization of Parameters in the Synthesis of Au NWs

In this work, MPA was selected as a protecting ligand, as well as a mild reductant. The highly monodispersed and atomically precise Au_4_ NCs without luminescence were prepared in the presence of excess MPA according to the previous report [42]. Upon the addition of Zn^2+^ ions, the highly ordered Au nanowires with strong luminescence were successfully obtained because Zn^2+^ could lead to the formation of superstructure by coordination interaction between the carboxylate group of MPA and Zn^2+^ ions. In order to obtain the Au NWs with excellent optical properties, the reaction conditions such as pH, Zn-to-Au ratio, temperature, and reaction time were explored because of strong assembling dependency on the synthesis conditions (Figure 1).

It was found that the pH had a great influence on the preparation of Au NWs due to the reaction of gold precursor and ligand-containing thiol group in controlled pH conditions. As shown in Figure 1A, the Au NWs prepared at pH 8.42 and 9.31 exhibited higher fluorescence intensity. However, Au NWs tended to form precipitates under pH 8.5. Hence, pH 9.31 was the optimum condition for the synthesis of Au NWs. The molar ratio of Zn-to-Au was another critical synthesis parameter. As shown in Figure 1B, a higher Zn-to-Au ratio (>6:1) could produce large complex precipitate while reducing the molar ratio (<3:1) formed product with less luminescence. At a Zn-to-Au ratio of 3.75:1, it was found that the obtained Au NWs exhibited high fluorescence quantum yield and excellent dispersion in water. Also, the photograph of Au NWs prepared at different Zn-to-Au ratio was displayed in Appendix A. In addition, the influence of the temperature and reaction time on fluorescence intensity was concurrently monitored. As shown in Figure 1C, weak fluorescence was observed after the introducing of Zn^2+^ ions for a few minutes and then increased gradually with time. It is also found that, not only the fluorescence intensity of Au NWs was increased, but also the time for reaching the maximum intensity was shortened when the temperature increased to 50 °C. At the temperature of 70 °C, the fluorescence firstly increased, and then sharply decreased, until it disappeared. Consequently, the optimal synthesis temperature and time for Au NWs preparation were 50 °C and 4 h, respectively.

### 3.2. Characterization and Properties of Au NWs

The morphology and structure of as-synthesized Au NWs were investigated by TEM and SEM (shown in Figure 2A,B). The TEM images revealed that the Au NWs were composed of uniform nanowires with a length of several hundred nanometers. Further, many nanowires were almost arranged along the same direction and formed the nanowire bundles. Consistent with the TEM result, SEM images also showed that the Au_4_ NCs actually arranged themselves into regular line-like architectures. FT-IR was further used as an effective tool to accurately interpret the self-assembly forces of Au NWs. As displayed in Figure 2C, the covalent affinity of Au to the thiol group of MPA resulted in the disappearance of the characteristic peak of thiol group at 2569 cm^−1^ on the Au NWs powder. The antisymmetric vibration (1702 cm^−1^) and symmetric vibration (1403 cm^−1^) of the carboxyl group changed to 1541 and 1388 cm^−1^ with the addition of Zn^2+^ ions, indicating the formation of a coordination bond between Zn^2+^ and carboxylate. Dynamic light scattering (DLS) showed the increase in average size of Au NWs with time passed during preparation process, and the overall size of Au NWs was approximately 50–200 nm after sufficient reaction time (displayed in Figure 2D). As shown in Appendix A, Au NWs were negatively charged with a surface zeta potential of ∼30 mV, implying that they might be well-dispersed in water. Compared with Figure 1C, it could be found that the fluorescence emission intensity was still increased when the Au NWs keep a similar size, which implied that the transformation of Au NWs from discrete aggregates to highly oriented assemblies is responsible for strong emission [39,43].

To better understand the optical properties of the as-prepared Au NWs, UV-vis absorption and fluorescence spectra characterizations were carried out (shown in Figure 3A,B). The UV-vis absorption spectra displayed three broad peaks at around 209, 345, and 477 nm, resulting from the discrete electronic structure of Au NWs. The inset of Figure 3A was the variation of optical absorption of self-assemblages as a function of time. Remarkably, Au NWs exhibited bluish-green emission centered at ~485 nm with large Stokes shift and wide excitation range, which is in accordance with the broad optical absorption. The half peak width was about 25 nm, which represented the high optical purity of highly ordered Au NWs. Moreover, the Au NWs exhibited excitation-independent fluorescence behavior when different excitation wavelengths were applied, suggesting that the fluorescence originated from homogeneously distributed emission sites. Taking the emission intensity at 485 nm into account, we chose 360 nm as the excitation wavelength in the following experiments. Photographs taken under visible light and UV light of Au_4_ NCs, Au NWs in water, and Au NWs powder were displayed in Figure 3C. We then evaluated the absolute QY of Au NWs fabricated under optimized conditions, the average absolute QY value of three as-formed Au NWs in water was 26.3%, and the lifetime was determined to be 7.96 ns (shown in Appendix A). Compared with the previously work [35,36,37,38,39,40,41], our synthesized Au NWs with regular self-assembled structure, high absolute QY, excellent water solubility, and economic ligand broadened their application in bioscience.

Further, pH-dependent fluorescence measurements of Au NWs between 5.0 and 10.0 in 10 mM Tris-HCl buffer solution were carried out. As exhibited in Figure 3D, an increase in fluorescence intensity of Au NWs was observed, with the pH changing in the acid condition. With pH varying from 7.0 to 8.0 at the interval of 0.1-pH unit, the fluorescence intensity was sharply decreased and changed in a linear fashion in the range from 7.0 to 7.8. And the fluorescence was almost entirely quenched at pH 8.0. These results indicated that Au NWs was much sensitive to pH in a basic environment and could be used as an indicator for monitoring pH-related bio-enzymatic reactions. Also, the stabilities of this fluorescent material were studied in this work. Appendix A depicted that the Au NWs exhibited good stability against continuous irradiation for 30 min, revealing the favorable anti-photobleaching of Au NWs. Besides, the Au NWs remain excellently stable, with a 13% and 30% increase in fluorescence intensity for one month, both at 4 °C and room temperature, respectively (Appendix A). This high QY, together with the regular morphology and superior stability of the self-assemblies, benefited the further application of Au NWs in biosensing fields.

### 3.3. Design of the Fluorescence Sensing System

Since the fluorescence intensity of Au NWs was found to correlate with the basicity of the environment, we speculated that a fluorescence signal-off nanoprobe based on the nanowire arrays could be proposed for monitoring the concentration of urea and urease (as shown in Figure 1) because urea can be specifically hydrolyzed to carbon dioxide and ammonia by urease catalyzed reaction, which can cause an obvious pH increase in the aqueous reaction medium resulting from ammonia generation. Thus, the fluorescence of pH-sensitive Au NWs was sharply quenched in the basic system. The fluorescence spectra of Au NWs sensing system in the presence or absence of target analytes were shown in Figure 4A. When the solution of Au NWs just mixed with urea or urease, respectively, no noticeable Au NWs fluorescence signal changes were observed, which implied that neither urea nor urease could effectively quench the fluorescence intensity of Au NWs. However, the fluorescence of Au NWs was remarkably quenched in the presence of both urea and urease due to the formation of ammonia. Thus, this fluorescent sensing platform can be utilized for both urea and urease detection based on the change of fluorescence emission intensity. Furthermore, we compared the influence of urea and ammonia on the luminescence intensity of Au NWs. It could be seen from Figure 4B that the presence of ammonia induced the quenching of Au NWs with a good linear relationship between F/F_0_ and ammonia concentrations in the range of 0–160 μM. However, just a little decrease of the luminescence intensity was observed after the addition of 500 μM urea. The above results indicated that the quenching reason was responsible for the presence of ammonia, further verifying the feasibility of our designed strategy for urea and urease detection.

The mechanism of pH-induced fluorescence quenching of Au NWs was described as follow. It is due to the disassembly of nanostructure of Au NWs or the formation of ground state complex without luminescence under alkaline conditions that caused the fluorescence quenching. The size of Au NWs (displayed in Figure 4C) decreased slightly until Au NWs solution become colorless and non-fluorescent with the increasing concentrations of ammonia, illustrating fluorescence decrease and not the disassembly of Au NWs. From Figure 4D, we could see that the fluorescence lifetimes of Au NWs and Au NW /ammonia system were not changed in aqueous solution. Furthermore, UV-via spectra of Au NWs in Figure 4E were evidently changed with varying concentration of ammonia. The lifetimes and UV-via spectra results prove that the main quenching is static in our proposed sensing system.

### 3.4. Urea or Urease Detection

The quantitative determination of urea and urease was evaluated under the optimum parameters. Figure 5A displayed that the fluorescence spectra of the Au NW/urease system towards urea with different concentrations. When the concentration of the enzyme urease was fixed at 12.0 U/L, the luminescence intensity of the sensing system was progressively decreased along with the urea concentration, increasing from 0–200 μM due to the enzymatic reaction causing a change in pH value. The fluorescence quenching degree F/F_0_ (where F and F_0_ were fluorescence intensity of the sensing system in the presence and absence of urea, respectively) showed good linearity against urea concentrations in the range of 0–100 μM (shown in Figure 5B). The LOD at a signal-to-noise (S/N) of 3 was 2.1 μM. For urease detection, 500 μM urea was used in the urease activity measurement. Figure 5C displayed the fluorescence spectra of the Au NW/urea system with and without urease. It can be clearly observed that the fluorescence intensity decreased proportionally with the concentration of urease ranging from 0 to 12 U/L. As plotted in Figure 5D, a good linear correlation could be obtained between the fluorescence signal ratio F/F_0_ (F_0_ and F were the luminescence intensity in the absence and presence of urease, respectively) and the concentrations of urease in the range of 0–12 U/L, and the detection limits of urease were estimated to be as low as 0.13 U/L. Compared with the previous reported urea and urease sensing methods in linear range and detection limit (Appendix A), it could be seen that our method shows excellent detection limit and linear range, which endowed our method with the ability of low urea and urease activity analysis and the advantage of less sample consumption.

### 3.5. Selectivity and Application in Real Samples

For a newly proposed nanoprobe, highly selective performance has been considered to be one of the greatest challenges for urea and urease detection in the complex sample analysis. The selectivity of the nanowire bundles with the same concentration of urea (500 μM) and urease (8 U/L) was implemented by adding potential interfering species, including metal ions and biological molecules. As depicted in Figure 6, negligible change on the fluorescence signal output of the sensing system was observed after exposure to the interfering compounds, confirming the excellent selectivity of the Au NW-based fluorescent probe toward urea and urease over other interfering species.

With excellent sensitivity and selectivity of this method, the proposed sensing system was used to determine the amount of urea and urease in human urine and serum samples, respectively. As Table 1 and Table 2 list, the obtained values of urea in human urines were calculated to be 203.7 and 137.9 mM by the proposed method. These results were close to 223.9 and 146.8 mM, which were measured by the urea-diacetylmonoxime reaction. The average recoveries of urea in urine and urease in serum reached 95.8–106.5% with a relative standard deviation (RSD) of less than 5%, which was acceptable for quantitative assays performed in biological media. All these data indicated this proposed fluorescent method has great potential applicability in the detection of urease activity and its catalytic substrate urea in biological samples.

## 4. Conclusions

In summary, novel water-soluble Au NWs with high quantum yield were successfully synthesized by an improved one-step heating method due to the coordination of Zn^2+^ ion with the carboxylate group in the gold-thiolate complexes. This Au NW emitted bluish-green fluorescence at 485 nm with a high absolute QY up to 32%. It was worth noting that the Au NWs possessed ultrasensitive pH-responsive property in the range of 7.0–7.8. Taking advantage of the nanowire performance, we constructed a simple sensing platform for urea and urease detection. Benefiting from the pH-sensitivity of Au NWs, our proposed fluorescent method showed sensitive detection of urea and urease with high selectivity and good linearities in the range of 0–100 μM and 0–12 U/L for urea and urease, respectively. The average recoveries of urea in urine and urease in serum were obtained from 95.8% to 106.5%, and the results of urea in primary urine was close to the urea-diacetylmonoxime method. Coupled with their optical properties, we believe that the Au NWs would be a promising candidate for applications in biological, medical, and pharmaceutical fields.

## Data Availability

Not applicable.

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
