# Peer review of "Highly Bright Gold Nanowires Arrays for Sensitive Detection of Urea and Urease"

_nanomaterials, 2022, doi:10.3390/nano12224023_

Round 1

Reviewer 1 Report

These nanowires do provide an interesting fluorescence dependence on pH, and I think their design is worth publishing, but I have some concerns with the results and some formatting corrections:

-          The final paragraph of the introduction contains too much results information.  This paragraph should instead describe the rough experimental approach and present the hypothesis of the paper, and leave the results for the results and conclusions sections.

-         -Section 2.3 – what buffers (or water) were used for the solutions?

-          Section 2.4 – How much is a lot of water?

-          Figure 2D is not referenced in the text

-          Figure 3B – it is difficult to tell the different spectra apart, perhaps use more contrating colours, or line styles to make it clearer

-          Section 3.5 – You state the results in the text as being in mM concentration, yet the table and previously stated linear range are both in μM.  Which is correct?

-          Section 3.5 – The measured concentrations of Urea in urine were well beyond the linear range of the test, how were these values accurately calculated?

-          How can you tell if the signal quenching is due to Urea or Urease presence since both are able to quench the NWs fluorescence?

-      The results section should include more comparison of the wires designed in this experiment to previously published works to help demonstrate their improvements

-          The conclusions need to include more of the specific results, as well as what experiments should come next

Author Response

  1. The final paragraph of the introduction contains too much results information. This paragraph should instead describe the rough experimental approach and present the hypothesis of the paper, and leave the results for the results and conclusions sections.

Response: Thanks for your kind suggestion. We have revised the content of this paragraph about experimental results at line 84-86, 88-90 and 91-96.

  1. Section 2.3 – what buffers (or water) were used for the solutions?

Response: I am so sorry for missing such important information. We have added the detection solution used in this work was water at line 141.

  1. Section 2.4 – How much is a lot of water?

Response: One of the human urines were collected from healthy volunteer after drinking about 2 L water at line 146.

  1. Figure 2D is not referenced in the text.

Response: We are sorry to make mistakes. Figure 2D was marked on relevant contents at line 198.

  1. Figure 3B – it is difficult to tell the different spectra apart, perhaps use more contrating colours, or line styles to make it clearer.

Response: Thanks for your good suggestion. We have revised Figure 3B.

  1. Section 3.5 – You state the results in the text as being in mM concentration, yet the table and previously stated linear range are both in μM. Which is correct?

Response: The concentration of urea in the primary urine was about mM concentration which was calculated by multiplying the detecting concentrations based on our method by the diluted times.

  1. Section 3.5 – The measured concentrations of Urea in urine were well beyond the linear range of the test, how were these values accurately calculated?

Response: The urine samples before measured were pre-treated by ultrafiltration and diluted to suitable concentrations, to ensure that the concentration of urea in diluted urine was within the linear range of our proposed method.

  1. How can you tell if the signal quenching is due to Urea or Urease presence since both are able to quench the NWs fluorescence?

Response: Thanks for your kind suggestion. Actually, as most of the common enzymatic catalytic biosensors, our reported fluorescent Au NWs biosensor was made function through refleting the reaction progress of urease catalytic reaction with change of fluorescence. Therefore, when used for quantitative detection, Au NWs biosensor can concurrently detect only one factor of the system while others will keep unchanged. Such signal quenching in this system can be only used for detecting urea or urease in one assay. On the other hand, urea and urease are not coexisted in normal human fluids. Although it has reported the existence of few urease, it is far lower than the limit of detection in our fluorescence assay. Therefore, such Au NWs biosensor can be also used for quantitively detecting urea in urine and urease in serum for real samples.    

  1. The results section should include more comparison of the wires designed in this experiment to previously published works to help demonstrate their improvements.

Response: We have added some previously published works at line 223-226.

  1. The conclusions need to include more of the specific results, as well as what experiments should come next.

Response: We have revised the conclusions of our manuscript at lines 352-356.

Reviewer 2 Report

The article presents a work to form suprastructures of aggregated gold clusters capped with carboxylate thiols that are linked with divalent cations of Zn2+. These structures can offer a fluorescent signal that depends on pH and temperature, so they are suitable for monitoring pH changes. The activity of urease, which yields ammonia, raises the pH and correlates the activity of urease. The article might be accepted after the authors remark the novelty of this work versus previous urea sensors based on urease.

Improvement recommendations:

Dynamic light scattering in figure 4c does not make sense, there is only one peak that probably matches with the length of the nanowires but there is no peak for the orthogonal distance, which should reflect the thickness of the nanowires. To ensure that the gold clusters are aggregating into nanowires instead of different shapes they should explain the lack of this signal.

The study lacks a comparison with current urea detection sensors or even biosensors. They can find some examples in ( doi.org/10.1016/j.snb.2008.04.025 ), but there are many others and more recent ones.

Over citation of Chinese authors (34 out of 43). It would be nice to broaden the literature beyond their boundaries.

Author Response

Reply to Reviewer #2:

  1. Dynamic light scattering in figure 4c does not make sense, there is only one peak that probably matches with the length of the nanowires but there is no peak for the orthogonal distance, which should reflect the thickness of the nanowires. To ensure that the gold clusters are aggregating into nanowires instead of different shapes they should explain the lack of this signal.

Response: Thanks for your kind suggestion. Just as your said, the value of dynamic light scattering results doesn’t reflect the real dimension of Au nanowires because of their anisotropic structures. The hydrodynamic dimeter from DLS measurement is also not the real length of nanowires. It can only well-reflect the dimension of globular structures. Even though, it can give us some information that the change of DLS results can wholly reflect the change of average nanowire dimensions. In our manuscript, the specific morphological information can be given from the transmission electron microscope (TEM) measurement.

  1. The study lacks a comparison with current urea detection sensors or even biosensors. They can find some examples in (doi.org/10.1016/j.snb.2008.04.025), but there are many others and more recent ones.

Response: Good suggestion. The comparison with current urea and urease sensors were displayed in Table S2 and 3 in the supporting information.

  1. Over citation of Chinese authors (34 out of 43). It would be nice to broaden the literature beyond their boundaries.

Response: Thanks for your good suggestion. We have revised the citation of our manuscript. And we will pay attention to this problem in our future work.
